# The Crossroads of Precision Medicine and Therapeutic Decision-Making: Use of an Analytical Computational Platform to Predict Response to Cancer Treatments

**DOI:** 10.3390/cancers12010166

**Published:** 2020-01-09

**Authors:** Amélie Boichard, Stephane B. Richard, Razelle Kurzrock

**Affiliations:** 1Center for Personalized Cancer Therapy, University of California Moores Cancer Center, La Jolla, CA 92093, USA; rkurzrock@ucsd.edu; 2CureMatch Inc., San Diego, CA 92121, USA; srichard@curematch.com

**Keywords:** precision medicine, neoplasms, molecular pathology, exceptional responders, therapeutic decision

## Abstract

Metastatic cancer is a medical challenge that has been historically resistant to treatments. One area of leverage in cancer care is the development of molecularly-driven combination therapies, offering the possibility to overcome resistance. The selection of optimized treatments based on the complex molecular features of a patient’s tumor may be rendered easier by using a computer-assisted program. We used the PreciGENE^®^ platform that uses multi-pathway molecular analysis to identify personalized therapeutic options. These options are ranked using a predictive score reflecting the degree to which a therapy or combination of therapies matches the patient’s biomarker profile. We searched PubMed from February 2010 to June 2017 for all patients described as exceptional responders who also had molecular data available. Altogether, 70 patients with cancer who had received 202 different treatment lines and who had responded (stable disease ≥12 months/partial or complete remission) to ≥1 regimen were curated. We demonstrate that an algorithm reflecting the degree to which patients were matched to the drugs administered correctly ranked the response to the regimens with a sensitivity of 84% and a specificity of 77%. The difference in matching score between successful and unsuccessful treatment lines was significant (median, 65% versus 0%, *p*-value <0.0001).

## 1. Introduction

Oncology is currently the largest area of clinical health expenditures for prescription drugs in the United States [1], and, based on the current pace of innovation in cancer care, anti-cancer treatment costs are only expected to increase [2]. Still, cancer remains a significant medical challenge, stubbornly resistant to conventional treatments. Further, many patients with cancer undergo multiple lines of treatment, with decreasing efficacy with each line. In addition to excessive healthcare costs, ineffective treatments unnecessarily expose patients to significant toxicities. 

One area of potential leverage in cancer care is the development of molecularly-driven combination therapies, offering the possibility of rational combinations of targeted ± immunotherapy agents that will overcome the appearance of treatment resistance [3,4]. Such an approach is consistent with the concept of “precision oncology,” which focuses on identifying drugs that are more likely to benefit the patients based on individual genetic, environmental, and lifestyle factors. The implementation of genome analyses in the clinical setting and concomitant development of specific targeted therapies has already led to improvements in the outcome of diseases previously described as incurable, such as metastatic melanoma or advanced colon cancer [5,6]. However, personalized oncology is still in its infancy, with only about 2% of cancer patients taking advantage of these services today [7]. The primary reasons for this slow adoption are because (1) the innate complexity of cancer biology renders the selection of treatment options difficult for oncologists and the current medical workforce is not prepared to analyze/utilize the results obtained from tumor profiling assays; and (2) the newest high-throughput profiling methods provide far too much information, rendering it difficult—even for a well-trained practitioner—to perform a deep review of the actionable knowledge for each of his/her patients in a timely manner [8]. Oncologists willing to practice personalized medicine are inundated by large amounts of data they are unsure how to use. Indeed, a recent survey demonstrated that 66% of oncologists desired genomic testing to guide treatment decisions, but 86% of them also acknowledged that more education is needed before it can be used wisely [9]. 

The focus of this study is to evaluate whether an advanced software platform using a personalized approach to oncology can predict the efficacy of systemic cancer therapies, and therefore assist physicians in their therapeutic decision-making process. The platform incorporates data from more than 230 Food and Drug Administration (FDA) approved drugs (mostly in oncology) and uses super-computing and scientific expertise in oncology, genomics, transcriptomics, proteomics, and cell biology to provide a ranking of customized combination therapies.

The accuracy of predictions presented by therapeutic decision-support platforms can be evaluated using exceptional responders reports. Exceptional responders are patients who present a unique positive response to treatments that are not effective for the majority of similarly treated patients. While the underpinnings of such responses remain unclear, it now appears that certain molecular features of the malignant tissue can be used to predict the efficacy of each regimen. If proficient, the platform should be able to discriminate, mainly based on the molecular fingerprint of each patient, regimens resulting in a positive outcome versus those associated with lack of response. In this study, a total of 70 cases and 202 regimens (including cytotoxic chemotherapies, targeted therapies, hormone therapies, and immunotherapies, used as a single agent or in combination) were curated and analyzed from the literature. We show that the platform was able to predict those regimens that would result in a response with a high degree of accuracy.

## 2. Results

The literature review retrieved 50 articles describing 70 patients, published between February 2010 and June 2017, describing cancer patients who were characterized as “exceptional” or “super-responders” to a particular line of treatment, and who also had a genomic, transcriptomic, and/or proteomic assay test performed on their tumor. The data on the drugs that led to the best response, as well as all prior therapies to which the patients had been resistant, as documented in the published case reports, were curated. The complete list of references and individual treatment lines reviewed is given in Appendix A.

The decision-support platform algorithm was applied to each treatment regimen in order to determine the predictive value of the system; the matching score reflects the effectiveness of each drug regimen against the pathogenic alterations observed within the tumor (molecular match), and a higher score should correlate with a better response.

A total of 70 patients with exceptional response to cancer treatment were curated. The patients had a median age of 57 years (range 29 to 84 years); the female to male ratio was 39/31. The tumor staging and/or TNM classification was available for 40 patients, 38/40 (95%) of them presented a locally invasive or metastatic disease (Appendix A). The histologic diagnoses were diverse (N = 23 different diagnoses), with the most frequent diagnoses being melanoma (N = 10, 14%), breast carcinoma (N = 6, 9%), and prostate carcinoma (N = 6, 9%). Rare cancer types were also included, e.g., PEComa (N = 3, 4%) or basal cell carcinoma (N = 2, 3%) (Figure 1).

Patients received an average of 3 treatment lines (ranging from 1 to 14 different regimens), and molecular profiles retrieved an average of 4 pathogenic alterations per tumor (Appendix A).

All 202 treatment lines described for these patients were further reviewed. These lines included single-agent regimens (N = 112, 55%) or combination therapies (N = 90, 45%; Table 1). The matching score, based on the molecular profile of the patient’s cancer, was retrospectively calculated by the computational platform for each treatment line and is represented in Figure 2, where the 70 exceptional responders are lined up on the X-axis (regardless of their cancer types) and the treatment regimens that each individual patient received are lined up on the Z-axis (and presented by descending matching scores). The matching score is represented on the Y-axis: the average matching score for all 202 treatment lines was 31%, ranging from 0% (corresponding to a regimen not matched to the molecular profile) to 100% (corresponding to a treatment matching all of the alterations presented by the molecular profile). The color of each column indicates the best response observed in the patient after administration of said treatment: 23/202 (11%) regimens resulted in a complete response, 47/202 (23%) regimens resulted in a partial response or a stable disease for more than 12 months; 53/202 (26%) regimens resulted in a stable disease for 6 to 12 months, and 79/202 (39%) regimens resulted in a progressive disease or a stable disease for less than 6 months (Table 1, Figure 2).

An example of matching scores calculated by the computational platform for a given patient is provided in Figure 3. Patient 51 is a 38-year old hormone-positive female breast cancer patient whose tumor presented a gene-copy loss of *PTEN*, as well as a gene-copy amplification of *CCND1*, *FGFR1*, and *PRKDC*. She received four regimens that were associated with progressive disease (FAC, paclitaxel, capecitabine, and tamoxifen) and one (anastrozole and everolimus combined) that led to a partial response lasting 11 months. Regimens and molecular profile corresponding to Patient 51 were entered into the decision-support platform and the resultant matching scores were computed: regimens that failed received a matching score of 17% (tamoxifen—matched to the ER positive expression) or 0% (FAC, cisplatin, capecitabine—not matched to any of the molecular markers presented by the tumor), while the regimen that led to the 11-month partial response received a score of 42% (anastrozole + everolimus—matched to the ER positive expression and the *PTEN* copy loss). To note, the computational platform was also able to suggest a novel regimen using three drugs approved by the FDA in oncology at the date of the analysis that would have fit the molecular profile presented by the patient with a matching score of 55%. 

Successful treatment lines are defined as regimens that lead to stable disease ≥12 months, complete response, or partial response. Seventy regimens were considered successful; 39 (56%) of these exceptional responses used single-agent regimens and 31 (44%) of them used combination therapies (Table 1). The average matching score for these successful lines was 60% (95% confidence interval (95% CI) = 52–68%), and the median score was 65% (Figure 4).

Amongst the 132 additional treatment lines described, 79 (60%) led to disease progression or disease stabilization for less than 6 months, and 53 (40%) led to disease stabilization for 6 to 12 months. Overall, 73 (55%) of these unsuccessful regimens used single agents and 59 (45%) of them used combination therapies (Table 1). The average matching score for these unsuccessful lines was 14% (95% CI = 10–19%), and the median score was 0% (Figure 4). The difference in median matching score between successful and unsuccessful treatment lines was highly significant (Mann–Whitney U = 1352, *p*-value <0.0001; Figure 4).

The algorithm performance and accuracy were then measured using indices usually defined as predictive biomarkers. The sensitivity, specificity, positive predictive value, and negative predictive value were estimated using the results obtained from the 202 treatment lines described above, and a threshold of 25% (optimizing the sensitivity and specificity criteria) was defined using the ROC curve method. The area under the curve (AUC) using the 25% threshold was estimated at 0.85 and is significant (*p*-value < 0.0001), therefore validating the discriminating utility of the matching score for the prediction of treatment outcomes (Figure 5). 

The sensitivity (or probability that the decision-support platform provides a score higher than 25% for patients who will actually benefit from the regimen received) is estimated at 84% (95% CI = 74–92%), and the specificity (or probability that the decision-support platform provides a score lower than 25% for patients who will not benefit from the regimen received) is estimated at 77% (95% CI = 69–84%; Table 2). The positive predictive value (the probability that a patient will respond well to a regimen when the matching score is higher than 25%) is estimated at 66% (95% CI = 59–73%), and the negative predictive value (the probability that a patient will not respond to a regimen when the matching score is lower than 25%) is estimated at 90% (95% CI = 84–94%; Table 2).

## 3. Discussion

Molecularly targeted cancer drugs are often developed with molecular diagnostics that attempt to identify which patients will achieve better outcomes on the new drug, and thus avoid innate treatment resistance. Such predictive biomarkers are playing an increasingly important role in precision oncology [10], but it is also recognized that the complexity of one patient’s tumor cannot be sufficiently impacted by the use of a single marker/single-agent method. 

Combination therapy has been widely used for the prevention or treatment of diabetes [11], Alzheimer’s disease, or cardiovascular diseases [12,13], and has proven to be particularly effective for the treatment of HIV/AIDS as well as cancers such as Hodgkin lymphoma [14,15,16]. Similarly, many expect personalized medicine and combination therapy to be the future of cancer care [17,18,19] versus scripted monotherapies in many cases [20,21]. Unfortunately, customized combination therapy is not widely explored in oncology [22], primarily because there is a plethora of possible mutations and associated drugs, which render the interpretation of each tumor molecular profile difficult [7,8]. 

In this analysis, we assessed the accuracy of a decision-support algorithm through the interrogation of all published case studies during the time period assessed that reported patients who were described as attaining an exceptional response to therapeutic regimens (and who also had available molecular results). This study demonstrates the high performance of such a matching score: the algorithm used by the decision-support platform correctly ranked the treatment response in 70 published cancer patients who received 202 different regimens and presented an exceptional response to at least one therapeutic regimen. The predictive score was significantly higher for treatments leading to a positive outcome in comparison to treatments that have failed (median, 65% versus 0%, *p*-value < 0.0001). Using a threshold score of 25%, the computational platform was able to categorize the regimens in positive or negative outcomes with a sensitivity of 84% and a specificity of 77%. Overall, 95% of successful treatment lines were assigned a score greater than 52%, whereas 95% of unsuccessful treatment lines were assigned a score lower than 19%. Factors beyond matching score or not yet incorporated in this version of the matching score may account for the few outliers that respond with low matching scores and vice versa. Indeed, a bias towards targeted therapies exists, as these drugs have been historically associated with the identification of specific molecular biomarkers. The matching score used in this study may, therefore, underestimate the potential activity of more conventional chemotherapies.

The study of “exceptional responders” is considered a promising research strategy and the subject of several ongoing investigations by cancer consortia, such as the NCI (National Cancer Institute, United States), that has embarked on the Exceptional Responders Initiative or UNICANCER (France), that has funded more than 1.2 M€ for the EXPRESS study (EXcePtional RESponSe—Exceptional and Unexpected Response to Targeted Therapies) [23,24,25]. While the underpinnings of such atypical responses remained unclear, it now appears that certain molecular features of the malignant tissue can be used to predict the efficacy of each regimen [26].

## 4. Methods

### 4.1. Therapeutic Decision-Support Platform Description 

The objective of a therapeutic decision-support platform is to rapidly analyze molecular and clinical contextual information, and then mine existing biology and medical evidence databases in order to suggest fully-customized comprehensive, personalized treatments for cancer patients. The software reviewed in this study (PreciGENE^TM^ therapeutic decision engine, CureMatch Inc., San Diego, CA, USA) uses genomic (DNA), transcriptomic (RNA), and/or proteomic (protein) results obtained from tumor solid tissue or blood-based molecular assays performed by Clinical Laboratory Improvement Amendments (CLIA)-certified third party companies. For each patient, the results of these assays are compared to several manually curated and continuously updated databases. First, the program identifies the functional impact of each unique molecular aberration observed and highlights the genes and/or pathways that can be pharmaceutically targeted (“actionable” alterations). Then, the platform uses a proprietary rules-based system to rank millions of combinations of drugs specifically tailored to the patient’s profile, taking into consideration the overall molecular fingerprints of the tumor, as well as specific activities and toxicities of the drugs considered. Drugs can be used alone or in combination, when relevant. Additional factors, such as the use of a minimal number of drugs, respect of FDA indications, and/or several cancer-consortium recommendations, are also considered. A report is generated, and several options are provided to the oncologist, who reviews the information and recommends a treatment plan for the patient.

### 4.2. Treatment and Molecular Profile Matching Score Evaluation

The therapeutic decision-support platform ranks all available personalized therapeutic options using a proprietary predicting algorithm called “matching score”. This score considers the fitness of one regimen against a molecular profile, as well as clinical criteria such as diagnosis, patient-specific drug-requirements (known allergies, accepted routes of administration, drug availability), and age (adult versus pediatric patients). The matching score is given as a percentage, where the highest score represents the better fit to the patient. This score is a derivative of previously described matching scores [17,27]. Known and clinically validated predictive biomarkers for the response to cytotoxic, targeted, hormone, or immunotherapy agents are considered. The effect of each drug on the alterations presented by the tumor is measured by a “hit score” and the effect of multiple drugs is added when used in combination. A drug is considered directly “matched” if its half-maximal inhibitory concentration (IC50) impacts the target at low nanomolar range (for small-molecule inhibitors) or if the target encodes the primary epitope recognized by an antibody. A drug is considered indirectly “matched” if the altered gene/protein is not directly targeted but otherwise related (i.e., located in the same signaling pathway) to the direct target of the drug. For a given combination, the matching score is calculated by dividing the number derived from the direct and indirect matches in the said combination (numerator) by the maximum “hit score” for the molecular profile considered (denominator), later finely tuned using a multi-parametric variable taking into account specific molecular and/or clinical criteria (expert-reviewed rule-based system). 

### 4.3. Exceptional Responders Case Study Retrieval

A comprehensive review of the medical literature retrieved reports of “exceptional responders” to cancer treatment published between February 2010 and June 2017. Exceptional responders are defined as cancer patients who achieved a dramatic and unexpected response to cancer treatments that are not effective for most other patients. The PubMed search was conducted using the keywords “exceptional responders” or “exceptional response” and “cancer”. Only those case studies comprised of sufficient tumor molecular profile description, details on the treatment regimen (monotherapy or combination therapy), associated response (defined using the RECIST (Response Evaluation Criteria in Solid Tumors) criteria, and duration of response (defined as time to treatment failure (TTF), in months) were kept for the analysis. 

### 4.4. Matching score Accuracy and Statistical Analysis

A total of 70 patients and 202 treatment lines were further analyzed: molecular and drug descriptions were entered into the decision-support informatics system that computed the corresponding predictive score, respecting the platform’s previously defined specifications and rule system. 

The matching score of each treatment line was then ranked and compared to the actual outcome observed in the patient. All regimens resulting in an exceptional response were considered as therapeutic successes (stable disease for ≥12 months or complete or partial response); other regimens were considered unsuccessful. Progression-free survival and overall survival could not be assessed because of a lack of reported data in these cases. Correlation between the predictive score and the clinical responses observed was evaluated using the Mann–Whitney U-test. The performances of the decision-support platform algorithm were later studied: high and low matching score categories were dichotomized using the receiver operating characteristic (ROC) curve method to examine its predictive accuracy on treatment outcomes. The threshold chosen was optimized for sensitivity and specificity criteria.

### 4.5. Ethics Approval and Consent to Participate

All cases used in this study were previously anonymized and published, and therefore informed consents have already been obtained by the authors/investigators of the previous study.

## 5. Conclusions

Although our study had certain limitations, including its retrospective nature and limited sample size, we show that the computational platform used is able to discriminate—on the sole basis of the molecular fingerprints described for each patient’s cancer—regimens that favor a positive versus negative response outcome. Combining data from multiple studies, as done here, may not be optimal and may lead to important biases, such as molecular profiling platform specificities, patient response inconsistencies, or missing criteria. The setup of clinical trials allowing the acquisition of large collections of normalized clinical and molecular descriptions of cancer patients in order to improve treatment efficacy predictions is warranted in the near future.

## Figures and Tables

**Figure 1 cancers-12-00166-f001:**
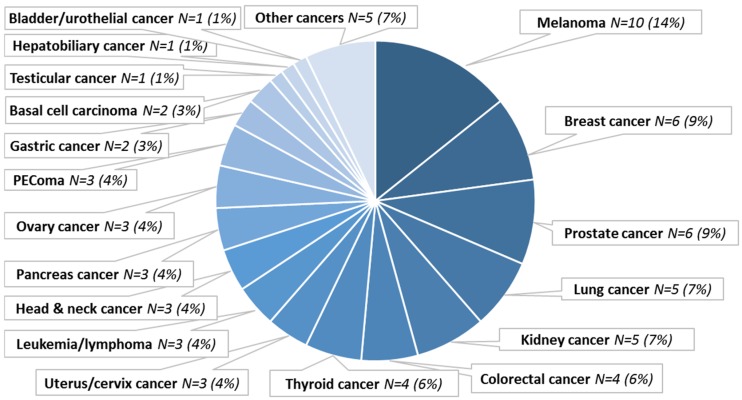
Cancer type distribution. Seventy patients with exceptional responses were curated. Abbreviations: N = number; PEComa = perivascular epithelioid cell tumor.

**Figure 2 cancers-12-00166-f002:**
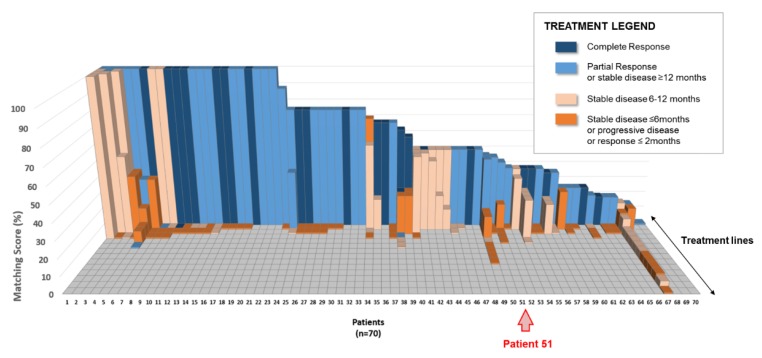
Matching score distribution of the 202 treatment regimens reviewed. Abbreviation: N = number.

**Figure 3 cancers-12-00166-f003:**
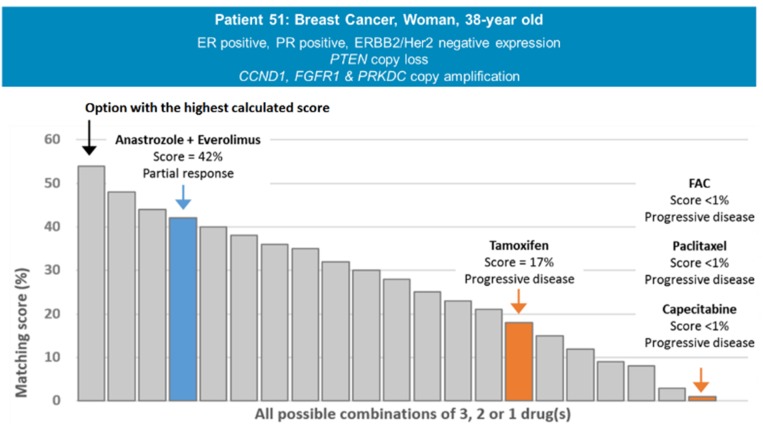
Comprehensive analysis of treatment regimens received by Patient 51. Abbreviations: *CCND1* = cyclin D1; ER = estrogen receptor; *ERBB2*/Her2 = erb-B2 receptor tyrosine kinase 2; FAC = fluorouracil, doxorubicin, cyclophosphamide; FDA = Food and Drug Administration; *FGFR1* = fibroblast growth factor receptor 1; PR = progesterone receptor; *PRKDC* = protein kinase, DNA-activated, catalytic polypeptide; *PTEN* = phosphatase and TENsin homolog.

**Figure 4 cancers-12-00166-f004:**
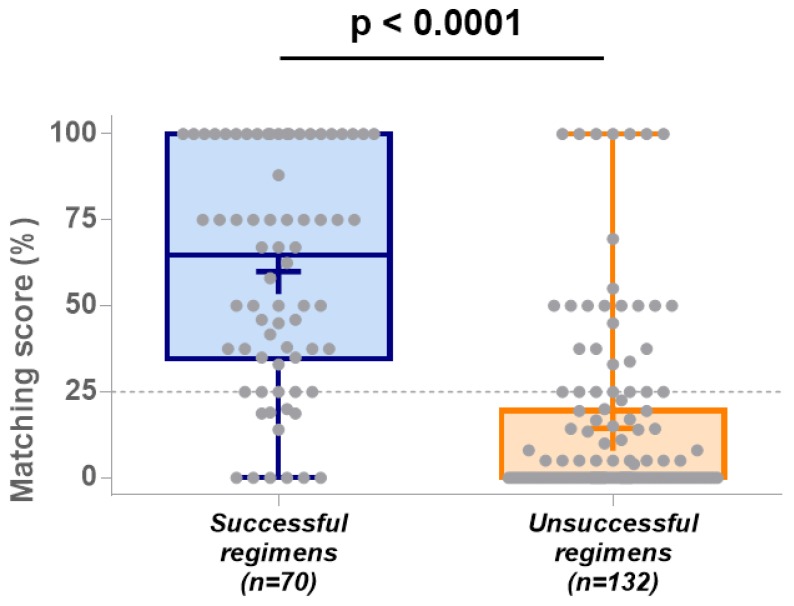
Matching score distribution between successful and unsuccessful regimens. Each regimen is represented by a grey dot; median, minimum and maximum scores are represented by a blue (for successful outcomes) or an orange (for unsuccessful outcomes) boxplot; average scores for both groups are indicated by a “+”. Abbreviation: n = number; *p* = *p*-value.

**Figure 5 cancers-12-00166-f005:**
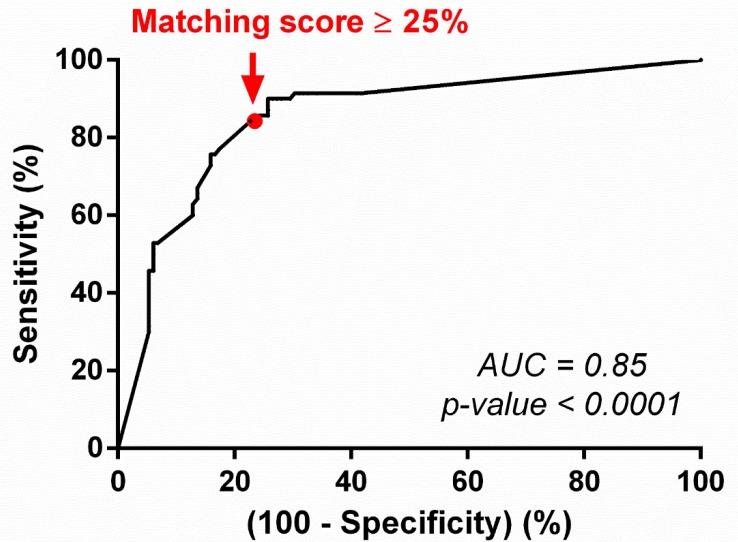
Operating characteristic (ROC) plot of the matching score for the prediction of clinical outcomes. Abbreviations: AUC = area under the curve; ROC = receiver operating characteristic.

**Table 1 cancers-12-00166-t001:** Description of the 202 treatment regimens reviewed.

Characteristics	Successful Regimens Reported * N (%)	Unsuccessful Regimens Reported ** N (%)
Total number of regimens	70 (35%)	132 (65%)
Single agent regimens	39 (56%)	73 (55%)
Combination regimens	31 (44%)	59 (45%)
Complete response	23 (33%)	0 (0%)
Partial response or stable diseass for more than 12 months	47 (67%)	0 (0%)
Stable disease for 6 to 12 months	0 (0%)	53 (40%)
Progressive disease or stable disease for less than 6 months	0 (0%)	79 (60%)

* Successful regimens indicate stable disease >12 months, partial response, or complete response. ** Unsuccessful regimens refer to any other outcome. Abbreviation: N = number.

**Table 2 cancers-12-00166-t002:** Evaluation of the decision-support platform algorithm performance *.

Characteristics	Successful Regimens Reported	Unsuccessful Regimens Reported
Predicted as favorable by the decision-support platform	59	30
Predicted as unfavorable by decision-support platform	11	102
Sensitivity ** (95% confidence interval)	84% (74–92%)	
Specificity ** (95% confidence interval)	77% (69–84%)	
Positive predictive value ** (95% confidence interval)	66% (59–73%)	
Negative predictive value ** (95% confidence interval)	90% (84–94%)	

* The performance evaluation of the decision-support platform algorithm is given using a threshold of 25% for the predictive score. ** The sensitivity/specificity measures the proportion of positive/negative outcomes that are correctly identified as such by the platform; the positive/negative predictive value measures the proportion of apparently-positive/negative scores for regimens that truly present a positive/negative outcome.

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
