# Peer review of "The Crossroads of Precision Medicine and Therapeutic Decision-Making: Use of an Analytical Computational Platform to Predict Response to Cancer Treatments"

_cancers, 2020, doi:10.3390/cancers12010166_

Round 1

Reviewer 1 Report

The authors present an interest paper regarding the management of cancer patients.

However, I suggest to improve:

the methodology section to indicate the website of the platform.

Author Response

We thank the reviewer for his/her examination.

The PreciGENE platform is a commercial tool available to medical doctors through licensing options with CureMatch Inc. This tool is therefore not directly available online.

More information may be obtained at https://www.curematch.com.

Reviewer 2 Report

The authors utilize a novel approach of using published data involving exceptional responders to assess their computational predictions.  This is an interesting premise and the review of exceptional responder articles is a good resource for other researchers. However, I think there are some fundamentally missing information for a complete study.

1) The data collected and used for training should gathered into a supplementary table and summarized into a figure like #1. Currently, we can't see if there are any biases in the groups of responsive treatments or resistant treatments (see point 2 for more details).

2) The authors say they used genomic, transcriptomic, AND/OR proteomic data. Does that mean if there is more data available, the potential for a match score will be higher? Again, it would help if the input data, treatments were provided. I realize there are proprietary algorithms involved, but some more data (unrelated to the algorithm) would help demonstrate the rigor of this study.

In the example for patient 51, it seems like the responsive treatment was everolimus which may have been targeting the PTEN copy loss. However, none of the resistant regimens are targeted drugs.  Thus it is possible the match score and exception responder cases are biased for targeted treatments. 

3) Combination of data from a multiple studies usually requires some data consolidation. Though the data are from CLIA labs, the assays all have different methods and nuances, and may not be directly comparable. This should be addressed in some manner, or minimally, the limitations of the approach should be discussed.

4) In the reviewed studies, are there molecular data for non-exceptional responders? Showing that these cases have a poor match score would serve as a type of negative control 

5) Figure 4. Would prefer to see boxplots with the dots rather than just a column graph showing the average value and CI. Boxplots display median and better match with a Mann Whitney U test. Also, the authors should discuss the discrepant cases where 100% match scores were for unsuccessful regimens and 0% match scores were for successful regimens.

6) I'm not sure why the matching scores for Patient 51 were re-calculated using only 5 treatments in Fig 3 (when they were already calculated in Fig 2 for 202 treatment lines). Then, in Fig 5, we are back to the 202 treatment lines for the AUC curve. I like the AUC curve idea for selecting a threshold, but it seems not to reflect the clinical scenario. For example for patient 51, the oncologist is likely considering between several breast cancer treatments and trials, and not thinking about thyroid, leukemia, melanoma, or other treatments (based on cancer types seen in Fig 1). 

7) The methods in 4.3 mention collecting RECIST criteria, which is a standard way of measuring successful response. The authors used SD>=12 months, CR, or PR as a successful treatment line, but should also evaluate based on RECIST. 

Author Response

We thank the reviewer for his/her interesting comments. We assessed each of the following points and modified the manuscript documents accordingly, whenever necessary.

1) The data collected and used for training should gathered into a supplementary table and summarized into a figure like #1. Currently, we can't see if there are any biases in the groups of responsive treatments or resistant treatments (see point 2 for more details).

A table reporting patient demographics, tumor types and treatment details has been included as Supporting Information Table 2.

2) The authors say they used genomic, transcriptomic, AND/OR proteomic data. Does that mean if there is more data available, the potential for a match score will be higher? Again, it would help if the input data, treatments were provided. I realize there are proprietary algorithms involved, but some more data (unrelated to the algorithm) would help demonstrate the rigor of this study.

Yes, it is possible that the potential for a matching score will be higher if more data is available.   More details about the potential of such matching scores is found in the paper from Sicklick et al.  (Nature Medicine, 2019), now included as reference #19.

In the example for patient 51, it seems like the responsive treatment was everolimus which may have been targeting the PTEN copy loss. However, none of the resistant regimens are targeted drugs. Thus it is possible the match score and exception responder cases are biased for targeted treatments.

The matching score is - de facto - biased towards already defined predictive biomarkers and corresponding pharmacology agents (either their mechanism of action is specific or not for a given target). We have added the following sentence to the Discussion (page 7):

Indeed, a bias towards targeted therapies exists, as these drugs have been historically associated with the identification of specific molecular biomarkers. The matching score used in this study may therefore underestimate the potential activity of more conventional chemotherapies.”.

3) Combination of data from a multiple studies usually requires some data consolidation. Though the data are from CLIA labs, the assays all have different methods and nuances, and may not be directly comparable. This should be addressed in some manner, or minimally, the limitations of the approach should be discussed.

The following statement has been added to the discussion (page 7):

Combining data from multiple studies, as done here, may not be optimal and may lead to important biases, such as molecular profiling platform specificities, patient response inconsistencies or missing criteria. The setup of clinical trials allowing the acquisition of large collections of normalized clinical and molecular descriptions of cancer patients in order to improve treatment efficacy predictions is warranted in the near future.”.

4) In the reviewed studies, are there molecular data for non-exceptional responders? Showing that these cases have a poor match score would serve as a type of negative control.

Only exceptional responders were included in this analysis, these patients being more often reported in the literature than regular responders or non-responders. Often, these patients received both treatment(s) that failed and successful treatment(s), allowing for comparison of the matching score accuracy in the same individuals.

To some extent, the treatment that failed before the exceptional responses reviewed in this analysis may be considered as negative controls for the matching score accuracy (each individual being his own control).

5) Figure 4. Would prefer to see boxplots with the dots rather than just a column graph showing the average value and CI. Boxplots display median and better match with a Mann Whitney U test.

Figure 4 has been modified accordingly. Boxplots representing minimum, median and maximum matching scores for successful (blue) and unsuccessful (orange) treatment lines have been added. Average scores for both groups are represented by a “+”.

Also, the authors should discuss the discrepant cases where 100% match scores were for unsuccessful regimens and 0% match scores were for successful regimens.

A statement acknowledging this limitation has been added to the discussion (page 7):

Factors beyond matching score or not yet incorporated in this version of the matching score may account for the few outliers that respond with low matching scores and vice versa.”.

6) I'm not sure why the matching scores for Patient 51 were re-calculated using only 5 treatments in Fig 3 (when they were already calculated in Fig 2 for 202 treatment lines). Then, in Fig 5, we are back to the 202 treatment lines for the AUC curve.

To clarify, Figure 3 represents in more details the results obtained for 1 patient (having received 5 treatment lines) found in Figure 2. The data used for both figures were originating from the same calculations.

I like the AUC curve idea for selecting a threshold, but it seems not to reflect the clinical scenario. For example for patient 51, the oncologist is likely considering between several breast cancer treatments and trials, and not thinking about thyroid, leukemia, melanoma, or other treatments (based on cancer types seen in Fig 1).

In the current clinical practice, the oncologist is more likely to consider treatments approved in the considered disease, unless the patient exhausted all predefined therapeutic options.

However, a decision-making algorithm such as the one presented here may also benefit from observations made in common disease types in order to offer new suggestions for rare and unconventional cancers.  Recently, a paradigm shift in biomarker-guided drug development has allowed the approval of 2 drugs based on their effect related to specific molecular aberrations independently from any tumor histology and anatomical location (namely pembrolizumab for microsatellite instability-high and mismatch-repair-deficient tumors and larotrectinib for neurotrophic receptor tyrosine kinase gene fusion-positive tumors).

7) The methods in 4.3 mention collecting RECIST criteria, which is a standard way of measuring successful response. The authors used SD>=12 months, CR, or PR as a successful treatment line, but should also evaluate based on RECIST.

Since this paper was performed using curation of data from the literature, we used the criteria reported, which were usually (but not always) RECIST criteria.

Reviewer 3 Report

In this manuscript Boichard et al. have evaluated the accuracy of PreciGENE platform to predict response to cancer treatment using pubmed published cases from 2010-2017 of exceptional or super-responders.

I think this topic is very novel but, in my opinion, there are relevant missing information and several points the authors need to address.

- The authors should mention that the same results were presented in ASCO 2018.

- The authors describe a platform to predict those regimens that would result in a response with a high degree of accuracy. Why there are only exceptional responders included in the analysis? Have you used a validation cohort? Have the authors used the platform with regular responders?

- In the introduction section, please modify the word single, not all the drugs do have a single mechanism of action.

- Please provide the patients’ information, diagnosis, tumor type and the regimens provided. There is no information regarding the drugs, which were used in monotherapy or in combination.

- The patients included were in advanced disease?

- Figure 1. Please include the number of patients in each tumor type.

- Please modify the figure legends. The information provided should be included within the text.

- Please explain more the results depicted in figure 2 and 4.

- In supporting information table 1 there are no 70 patients included. In “case number" column there are two numbers 24 and there is no number 4 and 33. Furthermore, the sum of the number of regimens is not 202.

Author Response

We thank the reviewer for his/her interesting suggestions. We assessed each of the following points and modified the manuscript documents accordingly, whenever necessary.

- The authors should mention that the same results were presented in ASCO 2018.

The following disclaimer has been added in page 10 of the main document:

“Prior Publication Disclaimer: The above results have been presented in a poster session of the American Society of Clinical Oncology (ASCO) Annual Meeting 2018.”

- The authors describe a platform to predict those regimens that would result in a response with a high degree of accuracy. Why there are only exceptional responders included in the analysis? Have you used a validation cohort? Have the authors used the platform with regular responders?

Only exceptional responders were included in this analysis, these patients being more often reported in the literature than regular responders or non-responders. Often, these patients received both treatment(s) that failed and successful treatment(s), allowing for comparison of the matching score accuracy in the same individuals.

The small number of available data (i.e. complete case reports including minimal patient and disease description plus molecular profiles and treatment descriptions) at the date of the analysis did not allow for an optimal validation cohort study.

The PreciGENE platform may be used for all type of cancer patients, at any time during their disease course, and is able to score all drug regimens. However, the matching score performance highly depends on the completeness of each patient description.

- In the introduction section, please modify the word single, not all the drugs do have a single mechanism of action.

In the introduction section (page 1) the following sentence:

“Still, cancer remains a significant medical challenge, stubbornly resistant to treatments that use a single mechanism of action.” has been replaced by “Still, cancer remains a significant medical challenge, stubbornly resistant to conventional treatments.”

- Please provide the patients’ information, diagnosis, tumor type and the regimens provided. There is no information regarding the drugs, which were used in monotherapy or in combination.

A table reporting patient demographics, tumor types and treatment details has been included as Supporting Information Table 2.

- The patients included were in advanced disease?

The stage and TNM classification were available for 40/70 patients: 36/40 patients presented a metastatic disease, 2/40 patients presented a locally invasive disease and 2/40 patients presented a tumor in situ (poorly differentiated thyroid cancer and high-grade endometrial cancer).

The Supporting Information Table 2 shows the details of all patients reviewed, and the following statement was added to the result section (page 2):

“The tumor staging and/or TNM classification was available for 40 patients, 38/40 (95%) of them presented a locally invasive or metastatic disease (Supporting Information Table 2).”

- Figure 1. Please include the number of patients in each tumor type.

The number of patients for each cancer type has been added on Figure 1 (page 3).

- Please modify the figure legends. The information provided should be included within the text.

The legends for Figure 1, Figure 2, Figure 3, Figure 4 and Figure 5 have been simplified. Corresponding descriptions have been included within the text in page 2, page 3, page 3, page 4, and page 6; respectively.

- Please explain more the results depicted in figure 2 and 4.

The descriptions corresponding to Figure 2 findings have been added in page 3:

The matching score, based on the molecular profile of the patient’s cancer, was retrospectively calculated by the computational platform for each treatment line and is represented in Figure 2, where the 70 exceptional responders are lined up on the X-axis (regardless of their cancer types) and treatment regimens that each individual patient received are lined up on the Z-axis (and presented by descending matching scores). The matching score is represented on the Y-axis: the average matching score for all 202 treatment lines was 31%, ranging from 0% (corresponding to a regimen not matched to the molecular profile) to 100% (corresponding to a treatment matching all of the alterations presented by the molecular profile). The color of each column indicates the best response observed in the patient after administration of said treatment: 23/202 (11%) regimens resulted in a complete response, 47/202 (23%) regimens resulted in a partial response or a stable disease for more than 12 months; 53/202 (26%) regimens resulted in a stable disease for 6 to 12 months, and 79/202 (39%) regimens resulted in a progressive disease or a stable disease for less than 6 months (Table 1 and Figure 2).”.

Figure 4 has been modified accordingly to reviewer #2 request.

The descriptions corresponding to Figure 4 findings can be found in page 4:

Seventy regimens were considered successful; 39 (56%) of these exceptional responses used single agent regimens and 31 (44%) of them used combination therapies (Table 1). The average matching score for these successful lines was 60% (95% confidence interval ([95% CI) ] = 52% to - 68%), and the median score was 65% (Figure 4).

Amongst the 132 additional treatment lines described, 79 (60%) led to disease progression or disease stabilization for less than 6 months and 53 (40%) led to disease stabilization for 6 to 12 months. Overall, 73 (55%) of these unsuccessful regimens used single agents and 59 (45%) of them used combination therapies (Table 1). The average matching score for these unsuccessful lines was 14% ([95% CI] = 10% to - 19%) , and the median score was 0% (Figure 4). The difference in median matching score between successful and unsuccessful treatment lines was highly significant (Mann-Whitney U = 1,352, p-value < 0.0001) (Figure 4).”.

- In supporting information table 1 there are no 70 patients included. In “case number" column there are two numbers 24 and there is no number 4 and 33. Furthermore, the sum of the number of regimens is not 202.

Two errors have been corrected in Supporting Information Table 1:

- Line 4, reference Garrido-Laguna et al. (2012)case number 33 (and not 24, as previously mentioned), 1 treatment line.

- Line 37, reference Gibson et al. (2017) – case number 4 (previously omitted), 4 treatment lines.

These corrections confirm the final numbers of 70 patients and 202 treatment lines analyzed.

Round 2

Reviewer 1 Report

I would like to thank the authors for improving their manuscript.

I find it much more clear due to the editing after revision.

Reviewer 3 Report

The authors have addressed all of my questions and, in my opinion, the manuscript has been improved. I have no additional comments.